# Transversality Conditions for Geodesics on the Statistical Manifold of Multivariate Gaussian Distributions

**DOI:** 10.3390/e24111698

**Published:** 2022-11-21

**Authors:** Trevor Herntier, Adrian M. Peter

**Affiliations:** Department of Computer Engineering and Sciences, Florida Institute of Technology, 150 W. University Blvd., Melbourne, FL 32940, USA

**Keywords:** geodesic, Fisher information, differential geometry, transversality, multivariate Gaussian

## Abstract

We consider the problem of finding the closest multivariate Gaussian distribution on a constraint surface of all Gaussian distributions to a given distribution. Previous research regarding geodesics on the multivariate Gaussian manifold has focused on finding closed-form, shortest-path distances between two fixed distributions on the manifold, often restricting the parameters to obtain the desired solution. We demonstrate how to employ the techniques of the calculus of variations with a variable endpoint to search for the closest distribution from a family of distributions generated via a constraint set on the parameter manifold. Furthermore, we examine the intermediate distributions along the learned geodesics which provide insight into uncertainty evolution along the paths. Empirical results elucidate our formulations, with visual illustrations concretely exhibiting dynamics of 1D and 2D Gaussian distributions.

## 1. Introduction

The importance of a metric to measure the similarity between two distributions has weaved itself into a plethora of applications. Fields concerning statistical inference [1,2,3], model selection [4,5,6] and machine learning have found it necessary to quantify the likeness of two distributions. A common approach to measure this similarity is to define a distance or divergence between distributions using the tenets of information geometry, e.g., the Fisher–Rao distance or the f-divergence [7], respectively. To the best of our knowledge, research and results in information geometry have predominantly focused on establishing similarities between two given distributions. Here, we consider an important class of problems where one or both endpoint distributions are not fixed, but instead, constrained to live on subset of the parameter manifold.

When one relaxes the fixed endpoint requirements, the development of finding the shortest path between a given distribution and constraint surface (not single distribution) must be reconsidered using transversality conditions [8,9] for the standard length-minimizing functional. This is precisely the focus of the present work, where we derive the transversality conditions for working in the Riemannian space of multivariate Gaussian distributions. This approach opens new avenues for research and application areas where one no longer needs to provide the ending distribution but rather a description of the constraint set where the most similar model must be discovered. For example, applications such as domain adaptation [10,11,12] can be reformulated such that the optimal target domain distribution is discovered among a constraint family starting from a known source distribution estimated from training data, or even model selection [4,5,6,13,14], where a search in a constrained family of distributions may be better aligned with the desired objective versus evaluating the MLE fit penalized by parameter cardinality.

In this work, we have purposefully chosen to work in the natural parameter space of multivariate Gaussians (mean vector, μ, and covariance matrix, Σ) and address the problem formulation in a completely Riemannian context. We are aware of the usual dually flat constructions [2,15,16] afforded by information geometry using divergence measures and the Legendre transformation. Though elegant in their algebraic constructions, these alternate parameterizations have yet to be employed in most statistical and machine-learning applications. Hence, we develop all geometric motivations and consequential mathematical derivations using Riemannian geometry and the natural parameter space, i.e., we employ the Fisher information matrix metric tensor and find the length-minimizing curve to the constraint set.

The remainder of this paper is organized as follows. In Section 2, we give a summary of important results concerning geodesics on manifolds. In Section 3, we provide a brief introduction to the techniques of calculus of variations with the goal of developing the Euler-Lagrange equations necessary to find the shortest path between two multivariate Gaussian distributions. In this section, we restrict ourselves to problems in which both the initial and final distributions are known exactly. Following this, in Section 3.3, we develop the conditions required to satisfy transversality constraints, or constraints where either the initial and/or final distribution are determined from those residing on a defined subsurface of the manifold rather than being exactly known. The results from these sections are employed in Section 4, where we explore various constraint surfaces and numerical experiments to demonstrate the utility of our variable-endpoint framework. Finally, some closing perspectives and remarks are given in Section 5.

## 2. Related Works

Most efforts towards measuring distances on statistical manifolds build on the foundation started by Fisher in [17], in which he introduces the idea of the information matrix. In [18] Kullback and Leibler published a pioneering effort to describe this distance. Works such as [19,20], endowed statistical distributions with geometrical properties. However, it was Rao [21] that expanded on the ideas of Fisher that defined a metric for statistical models based on the information matrix. The information matrix satisfies the conditions of a metric on a Riemannian statistical manifold, and is widely used because of its invariance [22]. This connection between distance and distributions encouraged others to explore the distance between specific families of distributions [23]. Among these families include special cases of the multivariate normal model [24], the negative binomial distribution [25], the gamma distribution [26,27], Poisson distribution [28], among others.

In [29], the authors offer a detailed exploration of geodesics on a multivariate Gaussian manifold. They show that there exists a geodesic connecting any two distributions on a Gaussian manifold. However, a closed-form solution for the most general case remains an open problem.

In [30] and expanded on in [31], the authors offer a very detailed discussion, focusing primarily on the univariate normal distribution for which a closed-form solution for the Fisher–Rao distance is known. Here, the authors focus on a geometrical approach, abandoning the “proposition-proof” format offered in previous research. With this geometric approach, closed-form solutions to various special cases are derived: univariate Gaussian distributions, isotropic Gaussian distributions, and Gaussian distributions with diagonal covariance matrix.

Another novel application of geodesics on a Gaussian statistical manifold is explored in [32], where the authors use information geometry for shape analysis. Shapes were represented using a *K*-component Gaussian Mixture Model, with the number of components being the same for each shape. With this, each shape occupied a unique point on a common statistical manifold. Upon mapping two shapes to their points on this manifold, the authors use an iterative approach to calculate the geodesic between these two points, with the length of the geodesic offering a measure of similarity of the shapes. Furthermore, because of the iterative approach to solving for the geodesic, all intermediate points along path are revealed. These points can be mapped to their own unique shapes, essentially showing the evolution from one shape to another. This shape deformation exhibits the benefit of analyzing more than just the distance between points on a manifold and that “walk” along the path has real substance.

In [33], the authors explore the complexity of Gaussian geodesic paths, with the ultimate goal of relating the complexity of a geodesic path on a manifold to the correlation of the variables labeling its macroscopic state. Specifically, the authors show that, if there is a dependence between the variables, the complexity of the geodesic path decreases. Complexity, for these purposes is defined as the volume of the manifold traversed by the geodesic connecting a known initial state to a future state, which is well defined. It is shown that this volume decays by a power law at a rate that is determined by the correlation between the variables on the Gaussian manifold.

In [34], the authors use the geometry of statistical manifolds to study how the quantum characteristics of a system are affected by its statistical properties. Similar to our work, the authors prescribe an initial distribution on the manifold of Gaussians and examine the geodesics emanating from it, without dictating a specific terminating distribution. The authors show that these paths tend to terminate at distributions that minimize Shannon entropy. However, unlike our work, these paths are free to roam on the manifold and are not required to terminate on a specific surface on the manifold. Furthermore, the most relevant part of the author’s work considers only univariate Gaussians with a two-dimensional parameter manifold, without ever considering higher dimensions.

Though we have chosen to work with Riemannian geometry, it is worth mentioning that information geometry often employs dualistic geometries that can be established using divergence measures. In [35], the authors detail the use of divergence measures to obtain the dual coordinates for space of multivariate Gaussians. However, they point out that the choice of divergence measure is not unique and resulting geometries lack the same interpretative power of the natural parameterization.

Though these previous works operate in the space of multivariate Gaussians and deriving geodesics therein, they all require defining the initial and terminal distributions on the manifold. In this work, we address a novel problem of finding the geodesics when the terminal conditions are hypersurface constraints rather than a single point. Technically, these transversality conditions are variable boundary conditions placed on the initial and final distributions requiring them to reside on a parametric surface typically defined by constraining the coordinates. The usefulness of these variable boundary conditions has emerged in many areas including physics [36] in which the author studied wetting phenomenon on rough surfaces and in [37], where the authors studied the elasticity of materials. Additionally, in [8,38,39], transversality conditions were employed in economic optimal control problems with a free-time terminal condition. However, as practical as transversality conditions have been in the above fields, their application in information geometry literature is deficient.

## 3. Geodesics for Fixed-Endpoint Problems

We begin by briefly developing standard calculus of variation results for discovering the shortest path between fixed points on a differentiable manifold. Then, the result is applied to the space of multivariate Gaussians, with detailed derivations for the case of bivariate Gaussians. We finally extend the formulation to show how to incorporate variable-endpoint boundary conditions.

### 3.1. General Euler-Lagrange Equations

Among differential calculus’ many applications are problems regarding finding the maxima and minima of functions. Analogously, techniques of calculus of variation operates on *functionals*, which are mappings from a space of functions to its underlying field of scalars. considering the functional L[y], the typical formulation of a calculation of variations problem is
(1)$$\begin{array}{cc} \min\; & L\left[y\right]=\int_{x_{0}}^{x_{1}}F\left(x,y,\dot{y}\right)dx\\ & y\left(x_{0}\right)=y_{0}\;y\left(x_{1}\right)=y_{1} \end{array}$$
where initial and terminal values are defined as ($x_{0},y_{0}$) and ($x_{1},y_{1}$), respectively and $\dot{y}$ is notation for the derivative. In general, *y* can be a vector of functions dependent on *x*, a vector of independent variables. The theory behind finding the extremum to problems such as these are analogous to single variable calculus, in which we a vanishing first derivative to locate critical points. Here, we locate the extremal functions using functional derivatives, leading to solving the Euler-Lagrange equations outlined in Section 3.2.

Though the formulation of finding the shortest path as a calculus of variations problem is rather elementary, its solution is involved. Obviously, in Euclidean geometry, this path is a straight line, and this distance is easily found. However, moving these ideas onto statistical manifolds complicates both the geometry and the calculus of this seemingly elementary problem. Analogous to the Euclidean setting, in a Riemannian manifold such as our space of Gaussians, solving for the shortest path *L*, involves the summation of many infinitely small arc lengths, ds
(2)$$ds^{2}={\dot{\theta }}^{T}g\left(\theta \right)\dot{\theta },$$
where θ is a parameter vector, g(θ) is a metric tensor dependent on the parameter vector and (${\cdot )}^{T}$ represents the transpose of a vector. The metric tensor for Euclidean space is the identity matrix but on the multivariate Gaussian manifold, this metric tensor is the Fisher information matrix, discussed later in Section 3.2.

This makes the functional we wish to minimize
(3)$$P=\int_{x_{0}}^{x_{1}}\sqrt{{\dot{\theta }}^{T}g\left(\theta \right)\dot{\theta }}dx$$
or, because the square root is a monotonically increasing function, we can conveniently use
(4)$$F=\int_{x_{0}}^{x_{1}}{\dot{\theta }}^{T}g\left(\theta \right)\dot{\theta }dx.$$
With this, the calculus of variation problem that solves for the minimum distance on a manifold is
(5)$$\begin{array}{cc} \min\; & F\left[\theta \right]=\int_{x_{0}}^{x_{1}}{\dot{\theta }}^{T}g\left(\theta \right)\dot{\theta }dx\\ & \theta \left(x_{0}\right)=\left[\theta_{01},\theta_{02},\ldots ,\theta_{0n}\right]\;\theta \left(x_{1}\right)=\left[\theta_{11},\theta_{12},\ldots ,\theta_{1n}\right] \end{array}$$ In the present context, the Fisher information metric tensor g(θ)=g(μ,Σ), the natural parameterization for multivariate Gaussians. Moreover, $\theta_{0}$ and $\theta_{1}$ are the parameters of the initial and final distributions.

The minimizer to Equation (Equation 5) is the well-known Euler-Lagrange equation, a system of second-order differential equations. These equations operate on the function in Equation (Equation 4). Accordingly, we define
(6)$$K={\dot{\theta }}^{T}g\left(\theta \right)\dot{\theta }$$ With this, the Euler-Lagrange equations are
(7)$$K_{\theta }-\frac{d}{dx}K_{\dot{\theta }}=0.$$
where $K_{\theta }$ and $K_{\dot{\theta }}$ are the functional derivatives with respect the curve θ(x).

### 3.2. Euler-Lagrange Equation for Gaussian Distributions

The Fisher information matrix is a measure of how much information about the parameter of interest from a multivariate distribution is revealed from random data space. Intuitively, it can be considered an indication of how “peaked” a distribution is around a parameter. If the distribution is sharply peaked, very few data points are required to locate it. As such, each data point carries a lot of information.

For a multivariate probability distribution, the Fisher information matrix is given by
(8)$$g_{i,j}\left(\theta \right)=\int f\left(x;\theta \right)\frac{\partial }{\partial \theta_{i}}\log f\left(x;\theta \right)\frac{\partial }{\partial \theta_{j}}\log f\left(x;\theta \right)dx,$$
where the index (i,j) represents the appropriate parameter pair of the multivariate parameter vector θ.

Alternatively, there are additional useful forms of the Fisher information, provided that certain regularity conditions are satisfied. First, the Fisher information matrix is the expectation of the Hessian of the log likelihood of the density function. Specifically,
(9)$$g_{i,j}\left(\theta \right)=-E\left[\frac{\partial^{2}}{\partial \theta_{i}\partial \theta_{j}}\log f\left(x;\theta \right)\right]=-E\left[H\right],$$
where *H* is the Hessian matrix of the log-likelihood.

Second, the Fisher information can be calculated from the variance of the score function
(10)$$g\left(\theta \right)=\mathrm{Var}(S_{f}\left(x;\theta \right)),$$
where
(11)$$S_{f}\left(x;\theta \right)=\nabla \log f\left(x;\theta \right).$$

Most importantly and for our purposes, the Fisher information matrix is the metric tensor that will define distances on Riemannian Gaussian manifold. Given a distribution on a manifold, by use of this metric tensor, we can minimize an appropriate functional to find a closest second distribution residing on a constrained subset the manifold. A class of problems covered by variable-endpoint conditions in the calculus of variations.

Consider the *n*-dimensional multivariate Gaussian with density given by
(12)$$f\left(x_{n}:\mu_{n},\Sigma \right)=2\pi^{-\frac{n}{2}}\det{\left(\Sigma \right)}^{-\frac{1}{2}}\exp-\frac{{\left(X-\mu \right)}^{T}\Sigma^{-1}\left(X-\mu \right)}{2}$$
where *X* is the random variable vector, $\mu =\mu_{1},\mu_{2},...,\mu_{n}$ is the *n*-dimensional mean vector of the distribution and Σ is the n×n covariance matrix.

Since the covariance matrix is symmetric, it contains $\frac{\left(n+1)(n\right)}{2}$ number of unique parameters, i.e., the number of diagonal and the upper (or lower) triangular elements. With the *n*-dimensional mean vector, the total number of scalar parameters in an *n*-dimensional multivariate Gaussian is $\frac{(n+3)n}{2}$, which will be the size of the Fisher information matrix. For all further developments in the parameter space, these parameters are collected in a single vector, θ such that
(13)$$\theta =\{\underset{\theta_{1},\theta_{2},...,\theta_{n}}{\mu_{1},\mu_{2},...\mu_{n}},\underset{\theta_{n+1},...,\theta_{\frac{(n+3)n}{2}}}{\sigma_{1,1}^{2},\sigma_{1,2}^{2},...,\sigma_{n,n}^{2}\}}$$

To clarify, this new parameter θ has the mean vector μ as its first *n* components and the resulting components are made up of the unique elements of the covariance matrix, starting with the first row, followed, by the second row but without the first entry, since $\Sigma_{1,2}=\Sigma_{2,1}$ and $\Sigma_{1,2}$ is already included in θ. We capture all the parameters of the multivariate Gaussian distribution in this non-traditional vector form because it is more conceptually in line with the calculation of the Fisher information matrix defined in Equation (Equation 8).

Therefore, using Equations (Equation 8) and (Equation 12), the Fisher information for the general multivariate Gaussian distribution is
(14)$$g_{ij}\left(\mu ,\Sigma \right)=\frac{1}{2}tr\left[\left(\Sigma^{-1}\frac{\partial \Sigma }{\partial \theta_{i}}\right)\left(\Sigma^{-1}\frac{\partial \Sigma }{\partial \theta_{j}}\right)\right]+{\frac{\partial \mu }{\partial \theta_{i}}}^{T}\Sigma^{-1}\frac{\partial \mu }{\partial \theta_{j}}$$
for which a very detailed proof can be found in the Appendix A of this paper. In the case of the bivariate Gaussian distribution, this 5×5 matrix has only 15 unique elements, because of its symmetry. Once again, the detailed derivation of each of the elements is provided in the Appendix A. The resulting metric tensor elements are
(15)$$\begin{array}{cc} g_{11} & =\frac{\sigma_{2}^{2}}{\sigma_{1}^{2}\sigma_{2}^{2}-\sigma_{12}^{2}}\\ g_{22} & =\frac{\sigma_{1}^{2}}{\sigma_{1}^{2}\sigma_{2}^{2}-\sigma_{12}^{2}}\\ g_{33} & =\frac{1}{2}{\left(\frac{\sigma_{2}^{2}}{\sigma_{1}^{2}\sigma_{2}^{2}-\sigma_{12}^{2}}\right)}^{2}\\ g_{44} & =\frac{1}{2}{\left(\frac{\sigma_{1}^{2}}{\sigma_{1}^{2}\sigma_{2}^{2}-\sigma_{12}^{2}}\right)}^{2}\\ g_{55} & =\frac{\sigma_{1}^{2}\sigma_{2}^{2}+\sigma_{12}^{2}}{{\left(\sigma_{1}^{2}\sigma_{2}^{2}-\sigma_{12}^{2}\right)}^{2}}\\ g_{12} & =-\frac{\sigma_{12}}{\sigma_{1}^{2}\sigma_{2}^{2}-\sigma_{12}^{2}}=g_{21}\\ g_{34} & =\frac{1}{2}{\left(\frac{\sigma_{12}}{\sigma_{1}^{2}\sigma_{2}^{2}-\sigma_{12}^{2}}\right)}^{2}=g_{43}\\ g_{35} & =-\frac{\sigma_{12}\sigma_{2}^{2}}{{\left(\sigma_{1}^{2}\sigma_{2}^{2}-\sigma_{12}^{2}\right)}^{2}}=g_{53}\\ g_{45} & =-\frac{\sigma_{12}\sigma_{1}^{2}}{{\left(\sigma_{1}^{2}\sigma_{2}^{2}-\sigma_{12}^{2}\right)}^{2}}=g_{54} \end{array}$$
All elements dealing with the information between a component of μ and an element of Σ vanish, which is a property extended to every multivariate Gaussian distribution of higher dimensions as well.

### 3.3. Variable-Endpoint Formulation: Transversality Boundary Conditions

Our development so far has focused on summarizing the usual situation of finding the length-minimizing path between two fixed points on a Riemannian manifold. We now shift to the situation of allowing one or both of the initial or final endpoints (distributions in this context) to be variable. This changes both the scope and the mathematics of the problem and requires the use of transversality boundary conditions.

Transversality conditions have been useful in other applications. In economics, the conditions are needed to solve the infinite horizon [40,41] problems. Other applications are found in biology [42] and physics [43]. However, to the best of our knowledge, there is little to no prior work investigating variable-endpoint formulations in the domain of information geometry. As we will demonstrate, the crux of employing these methods revolves around appropriately defining the parameter constraint surface. Though our results here provide concrete examples of interesting constraint surfaces, guidance on prescribing these subsets or developing techniques for automatically learning them will be important areas for future research.

The transversality conditions take into account the constrained hypersurface, ϕ(θ), from the coordinates parameterizing the distributions of the statistical manifold. If, for example, we are given an initial distribution and are asked to find which distribution on ϕ(θ) is closest, the usual geodesic problem formulated in Equation (Equation 5) now becomes
(16)$$\begin{array}{cc} \min\; & F\left[\theta \right]=\frac{1}{2}\int_{x_{0}}^{x_{1}}{\dot{\theta }}^{T}g\left(\theta \right)\dot{\theta }dx\\ & \theta \left(x_{0}\right)=\left[\theta_{01},\theta_{02},\ldots ,\theta_{0n}\right]\;\theta \left(x_{1}\right)=\phi \left(\theta \right) \end{array}$$ Therefore, in addition to satisfying the Euler-Lagrange equation, now with transversality conditions, the optimal solution must also satisfy
(17)$$\left[\begin{array}{c} K_{\dot{\theta_{1}}}\\ K_{\dot{\theta_{2}}}\\ \vdots \\ K_{\dot{\theta_{n}}} \end{array}\right]=\alpha \left[\begin{array}{c} \phi_{\theta_{1}}\\ \phi_{\theta_{2}}\\ \vdots \\ \phi_{\theta_{n}} \end{array}\right]$$

In Equation (Equation 17), the left-hand side is a vector tangent to the optimal path and the vector on the right right-hand side is the gradient of the terminal surface, which is orthogonal to the surface. Considering that the optimal path and the constraint surface intersect at the terminal distribution, this view of the transversality requirement implies that the tangent vector to the optimal path and the gradient of the constraint surface be collinear at the intersecting distribution. The scalar multiple α affects that magnitude of the vector and, from a geometric perspective, there is no loss of generality by setting α=1.

## 4. Results: Transversal Euler-Lagrange Equations for Bivariate Gaussian Distributions

We now turn our attention to various use cases of the transversal boundary conditions on the Gaussian manifold. We limit our derivations to bivariate Gaussians to make the calculations tractable and for visualization purposes. However, the same development is applicable to higher-dimensional Gaussians. It is worth mentioning that even in the fixed-endpoint scenario, there are no closed-form solutions for the geodesic when manifold coordinates include μ and Σ, with analytical solutions existing only for special cases such zero-mean distributions.

Using the Fisher information matrix defined in Equation (Equation 14), more specifically for the bivariate Gaussian distribution discussed in the Appendix, and employing the general form of the geodesic functional in Equation (Equation 4), we can define the integrand of the arc-length-minimizing function on the space of bivariate Gaussian distribution as
(18)$$\begin{array}{cc} K(\theta )= & \frac{\sigma_{2}^{2}{\dot{\mu_{1}}}^{2}}{k}+\frac{\sigma_{1}^{2}{\dot{\mu_{2}}}^{2}}{k}+\frac{{\left(\sigma_{2}^{2}\right)}^{2}{\left({\dot{\sigma }}_{1}^{2}\right)}^{2}}{2k^{2}}+\frac{{\left(\sigma_{1}^{2}\right)}^{2}{\left({\dot{\sigma }}_{2}^{2}\right)}^{2}}{2k^{2}}+\frac{\sigma_{1}^{2}\sigma_{2}^{2}{\dot{\sigma }}_{12}^{2}}{k^{2}}+\frac{\sigma_{12}^{2}{\dot{\sigma }}_{12}^{2}}{k^{2}}\\ & -\frac{2\sigma_{12}{\dot{\mu }}_{1}{\dot{\mu }}_{2}}{k}+\frac{\sigma_{12}^{2}{\dot{\sigma }}_{1}^{2}{\dot{\sigma_{2}}}^{2}}{k^{2}}-\frac{2\sigma_{12}\sigma_{2}^{2}{\dot{\sigma }}_{1}^{2}{\dot{\sigma }}_{12}}{k^{2}}-\frac{2\sigma_{12}\sigma_{1}^{2}{\dot{\sigma }}_{2}^{2}{\dot{\sigma }}_{12}}{k^{2}} \end{array}$$
where $k=\sigma_{1}^{2}\sigma_{2}^{2}-\sigma_{12}^{2}$.

We can use Equation (Equation 18) to derive the system of second-order differential equations, solutions to which yield the shortest path between two distributions.
(19)$${\ddot{\mu }}_{1}=\frac{{\dot{\mu }}_{1}{\dot{\sigma }}_{1}^{2}\sigma_{2}^{2}+{\dot{\mu }}_{2}\sigma_{1}^{2}{\dot{\sigma }}_{12}-{\dot{\mu }}_{2}{\dot{\sigma }}_{1}^{2}\sigma_{12}-{\dot{\mu }}_{1}\sigma_{12}{\dot{\sigma }}_{12}}{\sigma_{1}^{2}\sigma_{2}^{2}-\sigma_{12}^{2}}$$
(20)$${\ddot{\mu }}_{2}=\frac{{\dot{\mu }}_{2}\sigma_{1}^{2}{\dot{\sigma }}_{2}^{2}+{\dot{\mu }}_{1}\sigma_{2}^{2}{\dot{\sigma }}_{12}-{\dot{\mu }}_{1}{\dot{\sigma }}_{2}^{2}\sigma_{12}-{\dot{\mu }}_{2}\sigma_{12}{\dot{\sigma }}_{12}}{\sigma_{1}^{2}\sigma_{2}^{2}-\sigma_{12}^{2}}$$
(21)$${\ddot{\sigma }}_{1}^{2}=\frac{{\dot{\mu }}_{1}^{2}\sigma_{12}^{2}+{\dot{\sigma }}_{1}^{2}\sigma_{2}^{2}+\sigma_{1}^{2}{\dot{\sigma }}_{12}^{2}-{\dot{\mu }}_{1}^{2}\sigma_{1}^{2}\sigma_{2}^{2}-2{\dot{\sigma }}_{1}^{2}\sigma_{12}{\dot{\sigma }}_{12}}{\sigma_{1}^{2}\sigma_{2}^{2}-\sigma_{12}^{2}}$$
(22)$${\ddot{\sigma }}_{2}^{2}=\frac{{\dot{\mu_{2}}}^{2}\sigma_{12}^{2}+{\dot{\sigma }}_{2}^{2}\sigma_{1}^{2}+\sigma_{2}^{2}{\dot{\sigma }}_{12}^{2}-{\dot{\mu }}_{2}^{2}\sigma_{1}^{2}\sigma_{2}^{2}-2{\dot{\sigma }}_{2}^{2}\sigma_{12}{\dot{\sigma }}_{12}}{\sigma_{1}^{2}\sigma_{2}^{2}-\sigma_{12}^{2}}$$
(23)$${\ddot{\sigma }}_{12}=-\frac{\sigma_{12}{\dot{\sigma }}_{12}^{2}-{\dot{\mu }}_{2}\dot{\mu_{2}}\sigma_{12}^{2}-\sigma_{1}^{2}{\dot{\sigma }}_{2}^{2}{\dot{\sigma }}_{12}^{2}-{\dot{\sigma }}_{1}^{2}\sigma_{2}^{2}{\dot{\sigma }}_{12}^{2}+{\dot{\sigma }}_{1}^{2}{\dot{\sigma }}_{2}^{2}\sigma_{12}^{2}+{\dot{\mu }}_{2}\dot{\mu_{2}}\sigma_{1}^{2}\sigma_{2}^{2}}{\sigma_{1}^{2}\sigma_{2}^{2}-\sigma_{12}^{2}}$$

Along with satisfying this system of equations, the solutions presented here must satisfy transversality conditions at one or both the terminal and initial boundaries. In what follows, those conditions will be prescribed accordingly, considering various applications of interest.

### 4.1. Isotropic Terminal Distribution

Working with the full parameterization of a multivariate Gaussian distribution can be computationally expensive, especially when calculating their Fisher information matrix. For example, the number of parameters grows quadratically with dimension. However, isotropic Gaussian distributions, defined below in Equation (Equation 24) grow only linearly with the mean vector’s size, orders of magnitude more favorable. In addition to being computationally efficient, isotropic Gaussian distributions provide a submanifold prescribable by a tractable transversality condition. Accordingly, we formulate the following variable-endpoint problem: Given a general multivariate Gaussian distribution, what is the closest (in a geodesic sense) isotropic distribution?

Let $\Sigma_{i}$ be the covariance matrix of a multivariate Gaussian distribution with *n* mean components. This distribution is *isotropic* if
(24)$$\Sigma_{i}=\sigma^{2}I_{n}$$
where $I_{n}$ is the *n*-dimensional identity matrix.

Formally, let θ capture all the parameters of a multivariate Gaussian distribution according to Equation (Equation 13). The functional to minimize is given be
(25)$$\begin{array}{cc} \min\; & F\left[\theta \right]=\frac{1}{2}\int_{x_{0}}^{x_{1}}{\dot{\theta }}^{T}g\left(\theta \right)\dot{\theta }dx\\ & \theta_{0}=\left[\mu_{0},\Sigma_{0}\right]\;\theta_{1}=\phi \left(\mu_{1},\Sigma_{1}\right) \end{array}$$
where $\mu_{0}$ and $\Sigma_{0}$ are the known parameters of the starting distribution, but $\mu_{1}$ and $\Sigma_{1}$ are identified by solving the Euler-Lagrange equations while satisfying the transversal condition in Equation (Equation 24).

For the bivariate Gaussian distribution, the terminal surface described in Equation (Equation 24) can be defined by
(26)$$\Phi \left(\sigma_{1}^{2},\sigma_{2}^{2}\right)=\sigma_{1}^{2}-\sigma_{2}^{2}=0,$$
with $\mu_{1}$ free and $\sigma_{12}=0$. Appling Equation (Equation 17) to this surface, we obtain the condition
(27)$${\left(\sigma_{2}^{2}\right)}^{2}{\dot{\sigma_{1}}}^{2}+{\left(\sigma_{1}^{2}\right)}^{2}{\dot{\sigma_{2}}}^{2}+\sigma_{12}^{2}{\dot{\sigma_{2}}}^{2}+\sigma_{12}^{2}{\dot{\sigma_{1}}}^{2}-2\sigma_{12}\sigma_{1}^{2}\dot{\sigma_{12}}-2\sigma_{12}\sigma_{2}^{2}\dot{\sigma_{12}}=0.$$

Therefore, in addition to the Euler-Lagrange equations in Equation (Equation 19) through Equation (Equation 23), requiring the final distribution to be isotropic implies the geodesic must also satisfy Equation (Equation 27). Additionally, the terminal distribution must satisfy the conditions of constraint surface in Equation (Equation 26).

#### 4.1.1. Constant Mean Vector with Initial Isotropic Covariance

In this use case, we introduce a slight modification of the previous scenario, where now we set the mean vector of the initial and final distributions to be $\mu_{0}=\mu_{1}=\left[0,0\right]$. We are still interested in finding the closed isotropic Gaussian, but now not allowing the curve evolution to move the distribution’s mode.

For example, let us assume an initial distribution with
(28)$$\mu_{0}=\left[0,0\right],\;\Sigma_{0}=\left[\begin{array}{cc} 7 & 0\\ 0 & 2 \end{array}\right]$$
and the final distribution lie on the surface defined in Equation (Equation 26).

Applying the Euler-Lagrange equation with the transversality conditions to the problem above results in a final isotropic distribution with $\sigma^{2}=3.74$. Figure 1a shows the information path (dashed and curved) from the initial distribution to the chosen distribution on the isotropic constraint. A Euclidean path would end with a distribution with an isotropic variance equivalent to the average of the original variances. The geodesic path calculated under the Fisher information matrix is an indication of the curvature of the manifold in this region of the parameter space. It is possible in this special zero-mean case to analytically calculate the ending variance on the isotropic constraint surface. Given initial variances of $\sigma_{1}^{2},\sigma_{2}^{2}$, it can be shown that the final variance, $\sigma_{f}^{2}$ is given by
(29)$$\sigma_{f}^{2}=\sqrt{\sigma_{1}^{2}\sigma_{2}^{2}}.$$

In Figure 1b, we can see the evolution of all the parameters, starting from the initial distribution to end. Noteworthy is that, even though $\sigma_{12}$ is not required to stay at 0, there is no benefit for it deviating from 0, as seen in Figure 1b. The Fisher information matrix is independent of the mean vector and, since the values of the mean vector are also not part of our isotropic constraint on the final distribution, the mean vector is not compelled to change from the original distribution, justifying the exclusion of the mean vector’s path in Figure 1.

#### 4.1.2. Constant Mean Vector with Initial Full Covariance

In the example above, the distributions move along a 2-dimensional manifold parameterized just by the individual variances of the variables. Neither the mean vector, nor the off-diagonal values of the covariance matrices are considered by the Euler-Lagrange equation and therefore remain static. However, starting with a full covariance matrix changes the problem appreciably. For comparison, we will start with same diagonal elements of the covariance matrix but now incorporate the off-diagonal element.

The problem formulation is analogous to Equation (Equation 25) subject to the isotropic constraint in Equation (Equation 26). Furthermore, we define the initial distribution as
(30)$$\mu_{0}=\left[0,0\right],\;\Sigma_{0}=\left[\begin{array}{cc} 7 & -3\\ -3 & 2 \end{array}\right]$$

As seen in Figure 2, including $\sigma_{12}=-3$ alters where the geodesic terminates on the transversality constraint, with the final covariance matrix having diagonal elements of $\sigma_{1}^{2}=\sigma_{2}^{2}=2.24$. In Figure 2a, the effects of including the off-diagonal value $\sigma_{12}$ are clearly visible on the path of the geodesic causing it to deviate significantly. Figure 2b shows all values of the parameters along the geodesic. As mentioned before, the mean vector remains constant at all intermediate values of the distribution. Unlike before, the value for $\sigma_{12}$ must evolve to satisfy the terminal constraint surface, illustrated in Figure 2b and emphasized in Figure 2c.

#### 4.1.3. Starting on the Constraint Surface

When searching for the isotropic boundary condition, interesting paths occur if we start with an initial isotropic distribution, but require the mean vector to change. That is, if the initial distribution already resides on the terminal constraint surface, it would seem counter-intuitive if the geodesic is compelled to leave this constraint. However, as seen in Figure 3, this is exactly what happens.

Employing the same modeling equations as the previous example, we define the initial distribution as
(31)$$\mu_{0}=\left[-3,3\right],\;\Sigma_{0}=\left[\begin{array}{cc} 1 & 0\\ 0 & 1 \end{array}\right]$$
which is already isotropic. We search for the closest final distribution that is isotropic but with a mean vector $\mu_{1}=\left[3,-4\right]$. If we track evolution of just the mean vector, it starts in Quadrant II and moves to a distribution in Quadrant IV. Qualitatively, the initial and final distributions are geometrically symmetric. However, the isotropic uncertainty is considerably different.

The insights gleaned from this example shed new light into information evolution. Naively, one would think the shortest path would be one that maintains its current shape and just moves along the constraint surface to reach the desired μ. However, as shown in Figure 3, this is not the case. In fact, intermediate distributions obtain covariance matrices with $\sigma_{12}<0$, as demonstrated in Figure 3c. Instead of just staying on the constraint surface, the distributions stretch in the direction of the desired mean, which explains negative values of the covariance between the variables. This elongation of the covariance in the direction of final mean vector suggests that the information metric prefers uncertainty reduction in regions with few plausible solutions. Instead, the distributions along the geodesic “reach” for their destination, i.e., the initial mean vector in Quadrant II and final in Quadrant IV. This behavior is illustrated by the intermediate (red) ellipse in Figure 4.

To reiterate this insightful behavior of information flow, a second example was conducted with a mean vector that starts in Quadrant III, $\mu_{0}=\left[-3,-3\right]$, and seeks out a final mean vector in Quadrant I, $\mu_{1}=\left[3,4\right]$, as shown in Figure 5. The initial distribution is still isotropic and the requirement to end isotropic remains. However, as seen in Figure 6, the values of $\sigma_{12}$ acquire positive values along the geodesic. The path of the main diagonal variances remains unchanged.

### 4.2. Initial and Terminal Variable-Endpoint Conditions

It is possible to place transversality conditions on both the initial and final boundaries, with each being entirely independent of the other. Essentially, we are searching for a geodesic between two almost unknown distributions, with only minimal knowledge about the constraint set characterizing the initial and final hypersurfaces.

We consider the example where the final distribution is prescribed to be isotropic as before, but now the initial distribution must have a mean vector with equal components. This enforcement has the practical benefit of reducing the parameter dimensionality of allowable distributions. The problem is formulated as
(32)$$\begin{array}{cc} \min\; & F\left[\theta \right]=\frac{1}{2}\int_{x_{0}}^{x_{1}}{\dot{\theta }}^{T}g\left(\theta \right)\dot{\theta }dx\\ & \theta_{0}=\phi_{0}\left(\mu_{0},\Sigma_{0}\right),\;\theta_{1}=\phi_{1}\left(\mu_{1},\Sigma_{1}\right) \end{array}$$
where $\phi_{0}$ and $\phi_{1}$ represent the initial and final transversality constraint surfaces, such that
(33)$$\phi_{0}\left(\mu_{01},\mu_{02}\right)=\mu_{01}-\mu_{02}=0\;\mathrm{and}\;\phi_{1}\left(\sigma_{1}^{2},\sigma_{2}^{2}\right)=\sigma_{1}^{2}-\sigma_{2}^{2}=0.$$

To demonstrate this concretely, the initial distribution with unknown mean vector is prescribed with the following covariance matrix
(34)$$\Sigma_{0}=\left[\begin{array}{cc} 10 & 0\\ 0 & 2 \end{array}\right].$$
Similarly, the unknown isotropic terminal distribution is given the mean vector $\mu_{1}=\left[-3,13\right]$.

Using Equation (Equation 17), the constraint requiring the initial distribution to reside on $\phi_{0}$ imposes that the geodesic satisfy
(35)$$\left(\sigma_{2}^{2}-\sigma_{12}\right){\dot{\mu }}_{1}+\left(\sigma_{1}^{2}-\sigma_{12}\right){\dot{\mu }}_{2}=0.$$
As shown in Figure 7, the unknown initial mean vector satisfying the $\phi_{1}$ is $\mu_{0}=\left[7.4,7.4\right]$ and the final isotropic distribution has $\sigma_{1}^{2}=\sigma_{2}^{2}=26.4$. The behavior of the geodesic under these variable-endpoint conditions is shown in Figure 8.

## 5. Conclusions

In this work, we have explored new formulations for working on information geometric manifolds. Previous and contemporary work using the Riemannian geometry of statistical manifolds has focused on establishing geodesics between two fixed-endpoint distributions. Here, by employing techniques from the calculus of variations, we have developed constructions that allow variable endpoints that are prescribed by a constraint set rather than fixed points.

These transversality conditions on initial and final distributions, enable new insights into how information evolves under different constraint use cases. Though this present effort focused on just a small variety of constraints on Gaussian manifolds, this approach can be readily extended to other statistical families. This novel approach of relaxing fixed endpoints and moving constraint sets has the potential to impact several application domains that employ information geometric models. In future research, we plan to recast the problem of optimal distribution discovery, in areas such as model selection and domain adaptation, using the presented framework, which allows for greater expressive power. We also plan to investigate observational data scenarios where parameters must be estimated and impacts uncertainty modeling of the manifold parameters.

## Figures and Tables

**Figure 1 entropy-24-01698-f001:**
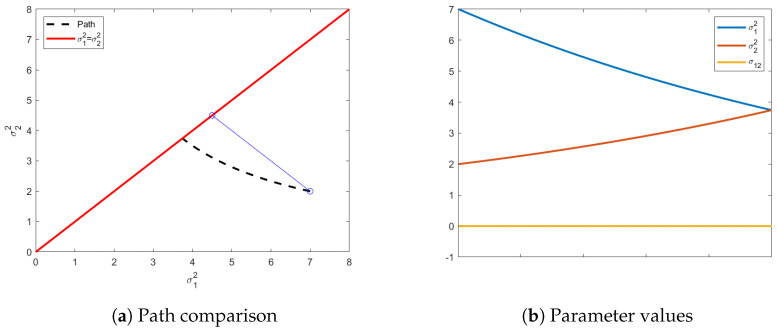
Shown above in (**a**) is the shortest path (dashed line) from a prescribed initial distribution with diagonal covariance matrix, $\sigma_{1}^{2}=7$, $\sigma_{2}^{2}=2$, to the closest isotropic distribution. The final distribution has $\sigma_{1}^{2}=\sigma_{2}^{2}=3.74$. The red solid line above is the transversality constraint $\sigma_{1}^{2}=\sigma_{2}^{2}$, represents the isotropic submanifold. Additionally illustrated, in blue, is the Euclidean path, which is clearly the straight-line path resulting from an identity metric tensor. In (**b**) are the paths showing the value of each element of Σ as the distributions move towards the transversality constraint.

**Figure 2 entropy-24-01698-f002:**
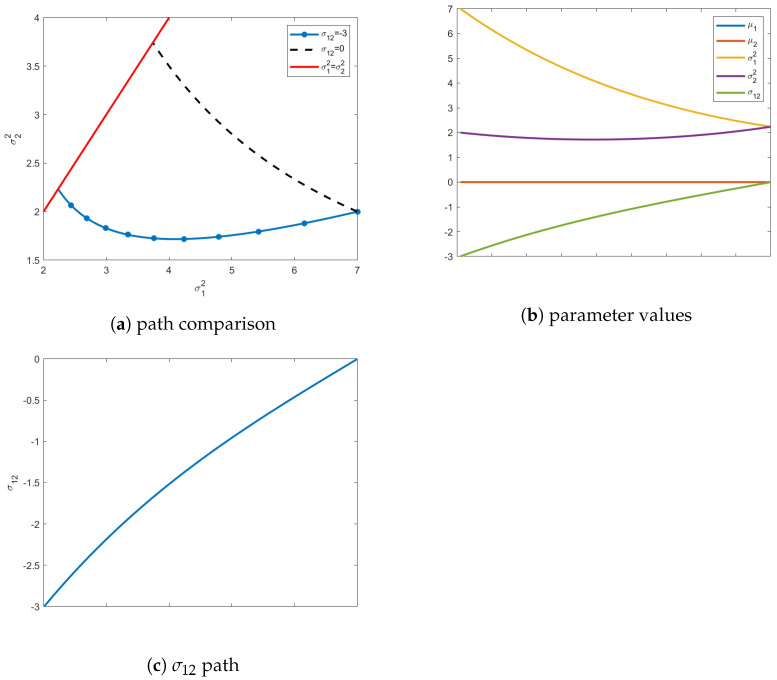
In (**a**) is the shortest path (blue line) from a prescribed initial distribution with an off-diagonal covariance element of $\sigma_{12}=-3$. Additionally, the path from (**a**) (dashed black) is shown to illustrate differences in the variances of the terminal distribution. The final distribution, when starting with a full covariance, has $\sigma_{1}^{2}=\sigma_{2}^{2}=2.24$. The red line above is the transversality constraint $\sigma_{1}^{2}=\sigma_{2}^{2}$, and represents the isotropic submanifold. Figure (**b**) captures the path of all parameters from the initial to the final distribution. Figure (**c**) highlights the values of $\sigma_{12}$ for each distribution in the geodesic on the manifold.

**Figure 3 entropy-24-01698-f003:**
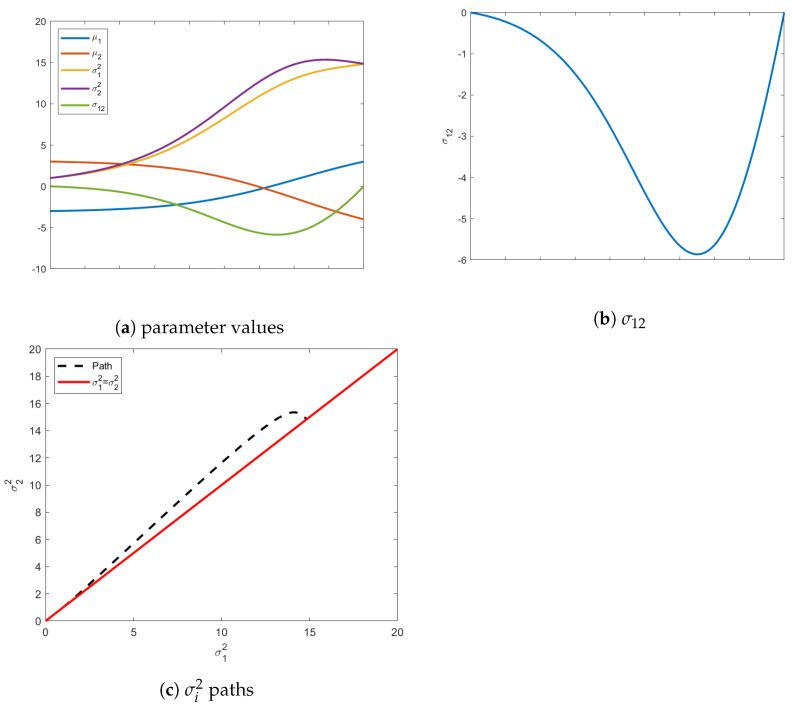
Above are evolutions of the geodesic from an initial isotropic distribution to a final isotropic distribution with a different mean vector. In (**a**), the values of all five parameters are shown at each iteration. Figure (**b**) highlights the values of $\sigma_{12}$ showing that it leaves the constraint surface and acquires negative values. The individual variances of the variables also temporarily abandon their required isotropicity as seen in (**c**). In (**c**), the dotted line shows the path of the $\sigma_{1}^{2}$ and $\sigma_{2}^{2}$ and the solid line shows the isotropic constraint surface.

**Figure 4 entropy-24-01698-f004:**
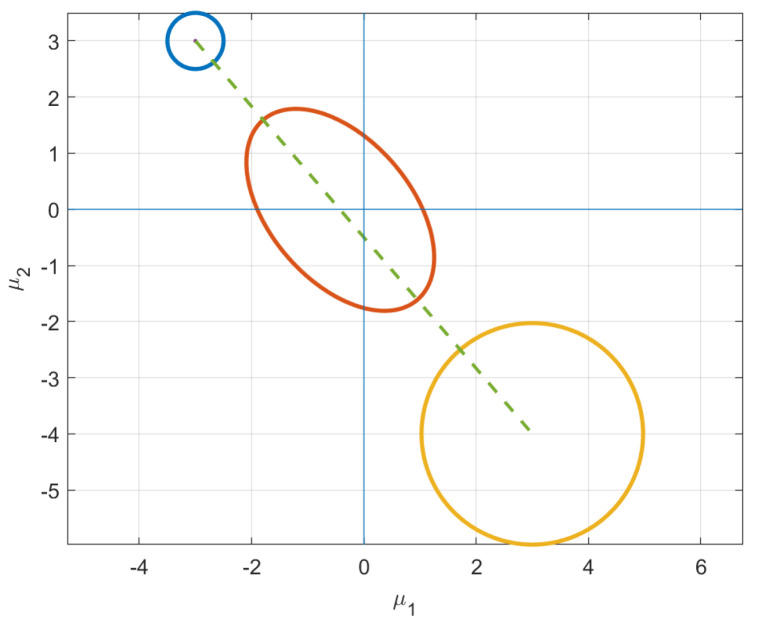
The ellipses above illustrate uncertainty contour for three different density functions along the geodesic. The initial distribution shown in the top left and final distribution in the bottom right are isotropic. The uncertainty evolution is clearly visible in the intermediate distribution which acquires negative covariance values as it “reaches” towards the final distribution.

**Figure 5 entropy-24-01698-f005:**
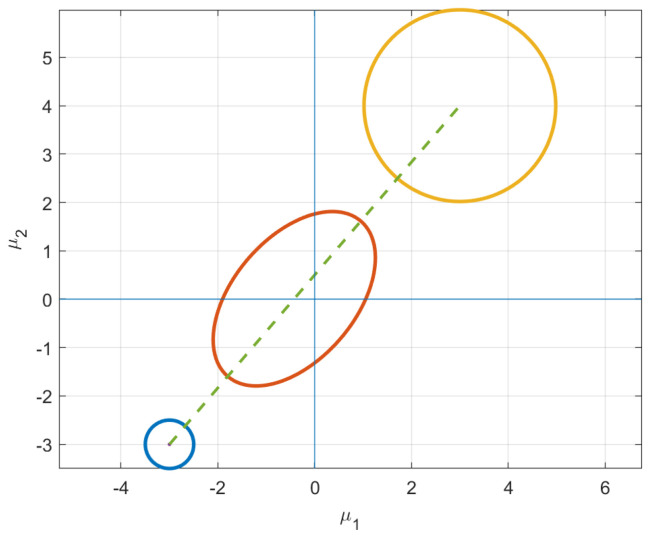
Similar to Figure 4, uncertainty level curves of three densities along the geodesic are shown. This time, the intermediate distribution acquires $\sigma_{12}>0$ along the geodesic as the distributions move from Quadrant III to Quadrant I along the dotted path.

**Figure 6 entropy-24-01698-f006:**
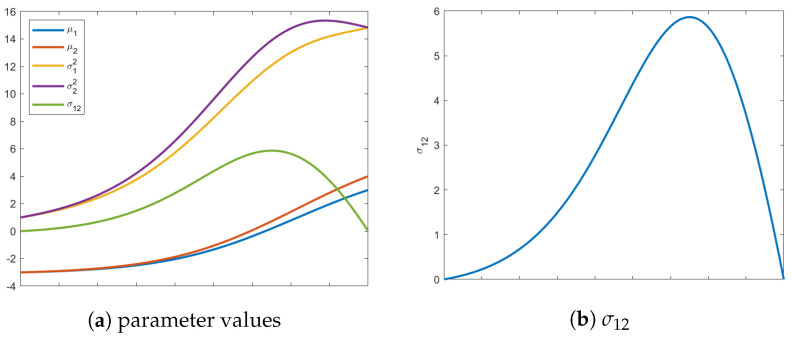
Illustration of geodesic paths from an initial isotropic distribution with a mean vector in Quadrant III and a final isotropic distribution with a mean vector Quadrant I. In (**a**), the evolutions of all five parameters are shown. Figure (**b**) highlights the values of $\sigma_{12}$ showing that it leaves the constraint surface and acquires positive values in contrast to the previous example. The individual variances of the variables also temporarily abandon their required isotropicity as seen in Figure 3c. In (**c**), the dotted line shows the path of the $\sigma_{1}^{2}$ and $\sigma_{2}^{2}$, with the solid red line representing the isotropic constraint surface.

**Figure 7 entropy-24-01698-f007:**
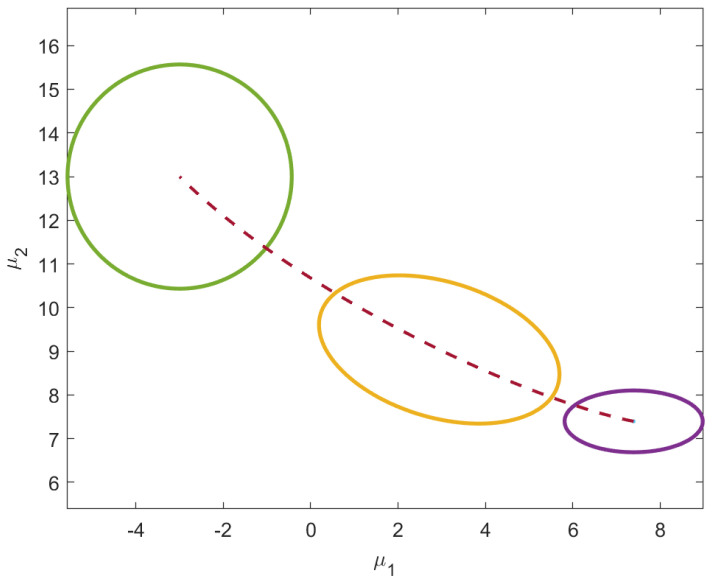
Alternate visualization of the resulting geodesic when both endpoints are allowed to vary. The uncertainty contour ellipses of the initial, intermediate and final distribution are also shown. The initial distribution represented by the bottom right ellipse, has components of the mean vector that are equal. The final distribution, at the top left of the path, isotropic.

**Figure 8 entropy-24-01698-f008:**
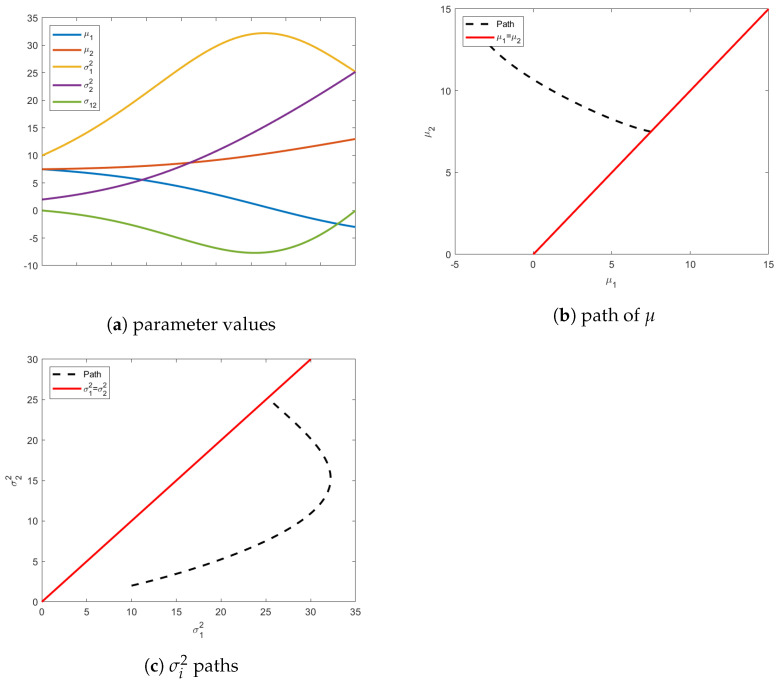
Illustration of geodesics resulting from initial and terminal variable-endpoint boundary conditions. Figure (**a**) shows the behavior of all parameters along the geodesic. Figure (**b**) shows how the geodesic (dashed) evolves the *initial* distribution to reach the constraint (solid) surface defined by the means. Similarly, Figure (**c**) shows the evolution of *final* distribution to the isotropic covariance constraint.

## Data Availability

Not applicable.

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
