# Peer review of "Transversality Conditions for Geodesics on the Statistical Manifold of Multivariate Gaussian Distributions"

_entropy, 2022, doi:10.3390/e24111698_

Round 1

Reviewer 1 Report

This paper needs careful review. As first there are many typing and English language errors.

Moreover, many times the expressions appear not rigorous from a mathematical point of view (for example often the concepts of geodesic distance and Riemannian metric seem to be confused with each other).

As for the theoretical contents, even if the basic idea is interesting in my opinion, it should be better focused. Indeed, the two approaches to the definition of geodesic are equivalent and well known in Differential Geometry literature.

Moreover, in the context of the statistical manifold of multivariate Gaussian distributions, also from the computational point of view, the proposed approach offers no advantages since, for n = 1 and n = 2, the analytical expressions of the geodesics are perfectly known.

What must therefore be emphasized is that the purposes of the research can lead to one or the other approach. For example, the knowledge of the exact form of the geodesic distance (which allows to quantify the diversity of two distributions) makes us lean towards the traditional approach if the aim is to cluster. On the contrary, the second approach is to be preferred if we search for the distribution, constrained to a submanifold, closest to a given one (model selection).  Furthermore, it is necessary to specify the research areas in which this latter approach is advantageous (for example in the FDA?)

Author Response

Please see that attached rebuttal.

Thanks for your sincere comments

Reviewer 2 Report

Please, read the file attached.

Author Response

Thanks for your sincere comments

Round 2

Reviewer 1 Report

The authors replied adequately and completely to all my perplexities and corrected all the critical issues of the first version of the paper.

Author Response

Thank you for the time spent on reviewing this paper

Reviewer 2 Report

In the reply report, the authors stated: "A comparison of our work and [2] has been added to the manuscript." However, I do not see in the body of the paper under consideration such a comparison.

In the end, I can suggest the publication on Entropy Journal once the authors have dealt with this issue.

Author Response

The following comparison has been added to the related works section.

In \cite{ciaglia2018}, the authors utilize the geometry of statistical manifolds to study how the quantum characteristics of a system are affected by its statistical properties.   Similar to our work, the authors prescribe an initial distribution on the manifold of Gaussians and examine the geodesics emanating from it, without dictating a specific terminating distribution.  The authors show that these paths tend to terminate at distributions that minimize Shannon entropy. However, unlike our work, these paths are free to roam on the manifold and are not required to terminate on a specific surface on the manifold.  Furthermore, the most relevant part of the author's work considers only univariate Gaussians, having a two-dimensional parameter manifold, without ever considering higher dimensions.

Round 3

Reviewer 2 Report

I would suggest the publication of the present version of the paper under consideration to Entropy Journal.